# The Genetic Landscape of Myelodysplastic Neoplasm Progression to Acute Myeloid Leukemia

**DOI:** 10.3390/ijms24065734

**Published:** 2023-03-17

**Authors:** Claudia Bănescu, Florin Tripon, Carmen Muntean

**Affiliations:** 1Center for Advanced Medical and Pharmaceutical Research, George Emil Palade University of Medicine, Pharmacy, Science and Technology of Targu Mures, 540142 Targu Mures, Romania; 2Genetics Department, George Emil Palade University of Medicine, Pharmacy, Science, and Technology of Targu Mures, 540142 Targu Mures, Romania; 3Pediatric Department, George Emil Palade University of Medicine, Pharmacy, Science, and Technology of Targu Mures, 540142 Targu Mures, Romania

**Keywords:** myelodysplastic neoplasm, progression, gene mutation, prognostic impact

## Abstract

Myelodysplastic neoplasm (MDS) represents a heterogeneous group of myeloid disorders that originate from the hematopoietic stem and progenitor cells that lead to the development of clonal hematopoiesis. MDS was characterized by an increased risk of transformation into acute myeloid leukemia (AML). In recent years, with the aid of next-generation sequencing (NGS), an increasing number of molecular aberrations were discovered, such as recurrent mutations in *FLT3*, *NPM1*, *DNMT3A*, *TP53*, *NRAS*, and *RUNX1* genes. During MDS progression to leukemia, the order of gene mutation acquisition is not random and is important when considering the prognostic impact. Moreover, the co-occurrence of certain gene mutations is not random; some of the combinations of gene mutations seem to have a high frequency (*ASXL1* and *U2AF1*), while the co-occurrence of mutations in splicing factor genes is rarely observed. Recent progress in the understanding of molecular events has led to MDS transformation into AML and unraveling the genetic signature has paved the way for developing novel targeted and personalized treatments. This article reviews the genetic abnormalities that increase the risk of MDS transformation to AML, and the impact of genetic changes on evolution. Selected therapies for MDS and MDS progression to AML are also discussed.

## 1. Introduction

Myelodysplastic neoplasm (MDS), previously well-known as the myelodysplastic syndrome, represents a heterogeneous group of myeloid diseases originating from the hematopoietic stem (HSC) and progenitor cells that lead to the development of clonal hematopoiesis [1,2]. 

MDS is characterized by a high risk of transformation into leukemia, namely acute myeloid leukemia (AML). MDS progression to AML appears in approximately 30–40% of MDS patients, is commonly named “secondary AML to MDS”, and is associated with a myeloblast count ≥20% [3,4]. MDS transformation into AML is relatively rare in low-risk MDS cases (LR-MDS) but is more frequent and rapid in high-risk MDS cases (HR-MDS) [5]. According to the 2016 revision of the World Health Organization (WHO) classification of myeloid neoplasms and acute leukemia, AML developing from MDS represents a distinct clinical and pathological entity that is included in the group of “AML with myelodysplasia-related changes (AML-MRC)”. The AML-MRC also includes patients with AML secondary to myeloproliferative neoplasms (MPNs), de novo AML with particular MDS-related cytogenetic abnormalities, and AML with multilineage dysplasia [6]. Based on the fifth WHO classification, myelodysplasia-related AML (AML-MR) was proposed for AML-MRC [1]. 

The International Prognostic Scoring System (IPSS), the most relevant MDS prognostic score, was adopted in clinical practice [7] but was proven to not be accurate for MDS cases who received treatment. IPSS analyzed cytogenetic abnormalities, the percentage of bone marrow blasts, and the number of cytopenias in order to establish the disease outcome for evolution to AML Due to the limitation of the IPSS (lack of inclusion of data regarding the degree of cytopenias and transfusion dependence, and the spectrum of dysplasia) a new IPSS score was proposed. 

The revised version of the International Prognostic Scoring System (IPSS-R) considers hematologic parameters (degree of cytopenia, dysplasia) as well as cytogenetic aberrations and allows for the risk stratification and risk-adapted therapy of MDS patients [8,9]. Gene mutations were proposed to be considered for the risk stratification of MDS cases for different studies [10,11]. The most recent Molecular International Prognostic Scoring System (IPSS-M) for myelodysplastic syndromes investigated 2957 samples from MDS cases from 24 centers, from which 234 samples were of cases with secondary/therapy-related MDS (s/t-MDS) [10]. An improved survival prediction accuracy for IPSS-M in ≥60 years old MDS cases was noticed [11]. Moreover, the IPSS-M led to improved prognostic accuracy in MDS patients with AML transformation, and by using the new molecular IPSS m where about 40% of MDS cases were re-stratified [10].

Moreover, a personalized prediction model (PPM-MDS) for the risk stratification of MDS patients was developed and validated. Even if the PPM-MDS model includes the seven most frequent genes that are mutated in MDS, it succeeded in realizing an improved prediction of survival and the risk of progression to leukemia [12]. An accurate prognosis of the patients will allow for appropriate treatment. For example, treatment with azacitidine plus lenalidomide may be associated with shorter survival in low-risk MDS patients, whereas the same therapy may positively influence survival in high-risk MDS cases treatment [12]. The characteristics of the IPSS, the revised version of IPPS, molecular IPSS, and a personalized prediction model for the risk stratification of MDS are presented in Table 1. 

Based on the fourth (2016) World Health Organization (WHO) classification of MDS, which integrated hematologic, morphologic, cytogenetic, and molecular genetic findings, MDS was classified as follows: MDS with single lineage dysplasia; MDS with multilineage dysplasia; MDS with ring sideroblasts; MDS with excess blasts (MDS-EB, subdivided into MDS-EB-1 and MDS-EB-2 according to the percentage of blasts in blood and bone marrow); MDS with isolated deletion del(5q), MDS, unclassifiable; refractory cytopenia of childhood [6].

The fifth WHO proposed the following classification of MDS: MDS with defining genetic abnormalities (MDS with isolated 5q deletion, with *SF3B1* mutation, and *TP53* biallelic mutation) and MDS, morphologically defined (include a distinct MDS type, namely hypoplastic MDS (cellularity < 20% in bone marrow) that responds to drugs used for the treatment of those patients (for example hypomethylating agents or possibly lenalidomide, etc.) [1,13]. The study performed by Zhang et al. investigated 852 cases diagnosed with MDS according to the fourth WHO criteria in order to assess the refinements of the fifth WHO classification. Thus, Zhang et al. re-classified MDS cases with *NPM1* mutation as AML and MDS unclassifiable (MDS-U) were considered clonal cytopenia of undetermined significance [13].

Recently, a new International Consensus Classification (ICC) of myeloid neoplasms and acute leukemias was proposed [14]. The ICC was developed in order to facilitate the diagnosis and prognosis of these disorders and to improve their treatment [14]. An unclassifiable MDS was not included in ICC. Thus, an MDS classification consisted of MDS with a *SF3B1* mutation (MDS-*SF3B1*, ≥10% variant allele frequency *SF3B1*), MDS with del(5q) (in fourth WHO classification known as isolated 5q deletion with up to one cytogenetic anomaly except for monosomy 7 or 7q deletion), MDS not otherwise specified (without dysplasia; with single lineage dysplasia, with multilineage dysplasia) which may associate any mutation except *TP53* or *SF3B1*; MDS with excess blasts (MDS-EB)(≥2% blasts in blood, and ≥5% blasts in bone marrow); MDS/AML (cytopenia, 10–19% blasts in blood and bone marrow, previously known as MDS with an excess of blasts > 10% in adults, MDS-EB2, may associate any cytogenetic aberration except AML defining, and any mutation except *TP53*, *CEBPA*, *NPM1* mutations) [14].

There are similarities between the fifth WHO classification and ICC regarding the MDS subtypes defined by genomic characteristics.

Chromosome anomalies and structural variants [for example, copy number alterations (CNA), translocations, inversions (inv), deletions (del), complex karyotype rearrangements, etc.] are found in MDS and also during transformation to AML [4].

By using sequencing techniques, recurrent somatic mutations in different genes were identified in MDS cases at diagnosis and during leukemic transformation. Recurrent mutations were found in genes implicated in RNA splicing (such as *SF3B1*, *SRSF2*, *U2AF1*, and *ZRSR2*), DNA methylation (*TET2*, *DNMT3A*, *IDH1/2*), genes involved in chromatin modification (*ASXL1*, *EZH2*), signal transduction (*JAK2*, *CBL*, *KRAS*), transcriptional regulation (*EVI1*, *RUNX1*, *GATA2*), and a cohesin complex [15,16]. It was indicated that the accumulation of epigenetic modifications represents an important factor in the MDS transformation to AML. Furthermore, it was demonstrated that *TET2*, *IDH1*, and *IDH2* gene mutations were driver mutations (that probably have a role in the pathogenesis of the disorder) and are acquired with the evolution of MDS to AML. It was estimated that the median time to develop leukemia from MDS was around seventeen months [17]. Different mutations tended to co-occur preferentially in the same patients, and it may define genomic subgroups of patients in the future. Thus, it became clear that the genetic investigation of MDS cases and identification of cytogenetic anomalies and gene mutations could have important prognostic significance and allow an improved prediction and prevention of leukemic transformation in MDS cases. Therefore, genetic anomalies in MDS and those described during MDS progression to AML are clinically relevant as prognostic markers, and patients carrying specific mutations could benefit from targeted or novel therapies for improving overall survival. 

While cytogenetics is important, it represents just one of several factors that affect the prognosis of MDS. The prognostic models that were developed rely on more variables, namely patient-related and disease-related, that include genetic factors that have an important impact on overall survival (OS) and the risk of leukemic transformation (chromosomal aberrations and gene mutations) (Figure 1).

In the present work, we aim to summarize the discoveries that help the understanding of MDS, including a state-of-the-art of diagnosis, risk stratification, prognostic scoring systems, and the risk-adapted treatment in order to improve the survival rate of affected patients and the prevention of its transformation into AML Considering that MDS is characterized by a high risk of transformation into leukemia it is essential to know the genetic anomalies that help clinicians for better clinical management. In this narrative review, we discuss the main evidence for this, taking into consideration the recent international recommendation, clinical trials, original research, reviews, our experience, and real-world evidence.

## 2. Cytogenetic of MDS Progression

According to the data reported by Papaemmanuil et al., cytogenetic anomalies were found in about 40–50% of MDS patients [16]; the most common chromosomal abnormalities are the loss of chromosome 7, deletions of the long arm of chromosomes 5 and 7, and gains of some chromosomes, such as chromosomes 8, 19, and 21 [18,19,20].

In MDS, the cytogenetic patterns are very heterogenous, and an important part of the patients acquire additional chromosomal abnormalities (ACA) as well [9]. The study realized by Jabbour et al. revealed that MDS patients that had ACA tended to have a higher risk of MDS transformation into AML or dying (HR = 2.02; *p* = 0.002) [21]. The presence of ACA represents a risk factor for AML transformation and is associated with an unfavorable prognosis leading to poor survival [21,22]. Previous research performed by Meggendorfer et al. showed that trisomy 8 appeared more frequently in MDS patients with progression to AML than in the control MDS group (13% vs. 3%; *p* = 0.015) [23], confirming that cytogenetic evolution in MDS is associated with disease progression to AML. Badar et al. analyzed 102 MDS cases and concluded that the acquisition of ACA did not influence overall survival (OS) at the time of MDS transformation into AML, even though the ACA was observed in 51% of patients from the moment of MDS diagnosis until the time of progression to leukemia [24]. The discrepancy between these studies may be due to the clinical heterogeneity that can be noticed in cases with similar chromosomal anomalies but with different somatic mutations. Cytogenetic aberrations that may occur in MDS cases and during MDS progression to AML are depicted in Figure 2.

Considering the importance of cytogenetics for the prognosis of MDS cases and also for choosing the most effective form of treatment, a Comprehensive Cytogenetic Scoring System for MDS and AML after MDS was proposed by Schanz et al. [22]. This cytogenetic risk classification in MDS included five different subgroups (very good, good, intermediate, poor, and very poor), unlike the International Prognostic Scoring System (IPSS), which includes three cytogenetic prognostic groups (good, intermediate, poor) [22]. 

The presence of the deletion of 7q (del7q) is associated with an intermediate prognosis, whereas the monosomy of chromosome 7 (−7 or loss of the whole chromosome 7) is associated with a poorer prognosis [9]. MDS cases with isolated del(7q) have longer overall survival (19 months) compared with patients that have isolated monosomy 7 (overall survival = 14 months) [25]. A higher risk of MDS transformation to AML was observed in cases with monosomy 7 compared with those with del(7q) [9,22]. Considering that monosomy 7 or deletion 7q are some of the most frequent cytogenetic aberrations to be identified in AML patients and is associated with poor OS, we may consider the presence of chromosome 7 anomalies to be an indicator of leukemic evolution. 

Other chromosomal anomalies such as inv(3)/t(3;3) or del(6p) were found in MDS patients that progressed to leukemia. Chromosomal abnormalities of chromosome 3 in cases with MDS were characterized by dismal outcomes (chemo-resistance and short OS).

A cytogenetic investigation is important in MDS considering the heterogeneity of achieved chromosomal abnormalities, such as isolated abnormalities, double abnormalities, or complex abnormal karyotypes (three or more abnormalities). Moreover, three distinct subgroups were proposed based on the presence of double abnormalities (defined as two different karyotype abnormalities identified in one cell), and they clearly distinguished different risks regarding OS as well as the risk of transformation into AML (poor risk group, including −7 or deletion 7q) [22]. 

Cytogenetic abnormalities, along with hematologic parameters and somatic mutations, were considered for an improved risk score of MDS patients and led to a clinical-molecular prognostic model (IPSS-M) [10]. Based on the IPSS-M score, MDS cases were classified into six survival groups (very low, low, moderately low, moderately high, high, and very high) [10]. The MDS entities were as follows: MDS with defining genetic aberrations [MDS with low blasts (<5%) and isolated deletion of chromosome 5q or del(5q) (MDS-5q); MDS with low blasts and *SF3B1* gene mutation (MDS-*SF3B1*); MDS with biallelic *TP53* mutation (MDS-bi*TP53*)] and MDS morphologically defined [MDS with low blasts, <5% marrow blasts (MDS-LB); MDS, hypoplastic; MDS with increased blasts (MDS-IB) that include MDS-IB1 with 5–9% marrow blasts; MDS-IB2 with 10–19% marrow blasts; MDS with fibrosis [1].

## 3. Molecular Signature

As specified above, nearly half of MDS cases had no chromosomal anomalies, suggesting that gene mutations were responsible for the pathogenesis of MDS. Myelodysplastic neoplasms are characterized by diverse somatic mutation patterns, with a frequency of 78–90% [9]. 

Goel et al. classified the mutations common in MDS and MDS progression to leukemia into the following main groups: spliceosome genes (*SF3B1*, *U2AF1*, *SRSF2*, *EZH2*), epigenetic modifiers (*TET2*, *DNMT3A*, *ASXL1*), transcription factors (*RUNX1*, *CEBPA*, *GATA2*), and cell signaling genes (*NRAS*, *KRAS*, *FLT3*) [17].

Recently, it was suggested that *SF3B1*, *SRSF2*, *ASXL1*, *TET2*, and *DNMT3A* gene mutations contribute to the risk of MDS evolution to leukemia and also influence therapy response and overall survival [19]. Considering the major role of genetic variants in the MDS progression to AML, we may assume that the progress of genomics and the implementation of genomic methods in clinical practice may improve clinicians’ capacity to predict leukemic transformation.

A large study that included 1019 patients from Germany suggested that the risk of MDS transformation to AML was driven by a solely genetic or epigenetic event [26]. Mutations in spliceosome genes and epigenetic modifiers are commonly observed in both MDS and secondary AML and occur early and contribute to MDS pathogenesis. Mutations that interest the genes that are involved in signal transduction (*JAK2*, *KRAS*, *NRAS*, *FLT3*, *CBL*) are secondary events [24,26]. Mutations of the genes implicated in epigenetic regulation (*IDH1*, *IDH2*, *BCOR*, *EZH2*) transcription factors and cell signaling are acquired during MDS progression [27]. Different studies reported that genes implicated in epigenetic regulation and splicing are the most frequently interested in mutations in MDS cases [28,29,30]. 

According to the descending order of the frequency of mutations, these interest the following genes: *SF3B1*, *TET2*, *ASXL1*, *SRSF2*, *DNMT3A*, *RUNX1*, *U2AF1*, *ZRSR2*, *STAG2*, *TP53*, *NRAS*, and *EZH2* [28]. 

*SF3B1* gene mutations confer an improved clinical outcome in patients, while *TP53*, *RUNX1*, *ASXL1*, *ETV6*, and *EZH2* are predictors of worse outcomes and are associated with inferior OS in MDS cases independent of other recognized risk factors [31].

It was shown that mutations that interest genes encoding epigenetic modifiers, mainly *EZH2*, *ASXL1*, *SETBP1*, *BCOR*, and *IDH2*, confer adverse prognoses in MDS [28,32]. Additionally, mutations that interest the *TP53* gene, transcription factors, and signal transduction initiators such as *NRAS*, *KRAS*, *NF1*, *JAK2*, *CBL*, and *FLT3* are associated with an unfavorable prognostic risk in MDS [10,28]. The imbalance between apoptosis and proliferation may lead to clonal expansion and explain MDS transformation. A “two-hit” model was proposed for MDS progression to AML [(first-hit: mutations in genes affecting the differentiation of cells (*TET2*, *RUNX1*) followed by the second-hit in the genes that influence the proliferation and survival of cells (*FLT3*, *NPM1*, *IDH1*, *IDH2*)] [33].

MDS progression to AML is driven by clonal evolution (known as the development or expansion of a subclone with a distinctive set of gene mutations) and is associated with the acquisition of novel driver variants [4,16,34]. The MDS transformation’s rate into AML increased as the number of driver mutations increased (*p* = 0.0001) [16].

It was reported that the mutations that interested the transcription factors (e.g., *RUNX1*, *CEBPA*, *GATA2*) and activating signaling genes (e.g., *FLT3*, *RAS* family genes) were more common in MDS evolution to AML, considering that these variants were attained later during disorder progression in a subgroup of cells that extended [4].

The study of Reinig et al. noticed that the most frequent mutated gene was *RUNX1* (28% of cases), followed by mutations in *U2AF1*, *SRSF2*, and *NPM1*, all with a similar frequency of 17% [31]. Mutations of the *RUNX1* gene were significantly associated with MDS transformation into AML compared with non-transformation in AML cases [31]. Additionally, it was observed that *NPM1* gene mutations were more frequently identified in cases with MDS transformation to AML than in cases diagnosed with MDS (*p* < 0.02) [31]. Liu et al. suggested certain patterns in the combination of gene mutations in patients with AML transformation from MDS [35].

In this paper, we overview the molecular basis of progression from myelodysplastic syndromes to AML as advances in genomics have unraveled particular gene mutations that are important predictors of prognosis and leukemic transformation. The order of gene mutation acquisition is not random [4], and a specific order of mutation acquisition was observed (Figure 3). Mutations in the genes implicated in various cellular pathways were identified in most MDS cases and during disease evolution [36] and will be discussed as follows: RNA-splicing factors (for example *SF3B1*, *ZRSR2*, *SRSF2*, *U2AF1*), epigenetic regulators (such as *DNMT3A*, *TET2*, *IDH1*, *IDH2*, *ASXL1*, and *EZH2*), transcription factors (for example *RUNX1*, *ETV6*, *GATA2*, *BCOR*), cell-cycle regulators (for example, *TP53*, *CDKN2A*), cell-signaling molecules (for example, *NRAS*, *KRAS*, *PTPN11*, *CBL*, *JAK2*, *FLT3*), as well as other mutations (for example in *NPM1* gene).

### 3.1. RNA Splicing Mutations

The alternative splicing of pre-mRNA represents one of the most frequently dysregulated processes in cancer. Mutations in genes (*SRSF2*, *SF3B1*, *ZRSR2*, *U2AF1*, *ZXRSR2*, *SF1*, and *SF3A1*) that encode the spliceosomal proteins are the most frequent recurrent mutations in MDS cases [37,38] identified in 30.1% of cases [31].

Spliceosome mutations are more frequent (39%) in cases with AML transformation from MDS [31,39]; this may be explained by the fact that spliceosome gene mutations have been correlated with the presence of mutations that interest the genes involved in the regulation of the cell cycle and proliferation that contribute to MDS pathogenesis. It was reported that spliceosome mutations are founding genetic alterations and are usually mutually exclusive to each other [40], but the presence of more spliceosome mutations in MDS patients is rarely observed [41]. The presence of mutations that interest the spliceosome genes (e.g., *SF3B1*, *SRSF2*, *U2AF1* genes) suggests AML progression from MDS, even in patients with a negative history of MDS diagnosis [4]. In MDS, spliceosome mutations occur commonly in SRSF2, SF3B1, ZRSR2, and U2AF1 genes, while *SF3A1*, *SF1*, and *ZXRSR2* gene mutations are rare and were reported with a frequency of 1% for each of them.

#### 3.1.1. SF3B1 (Splicing Factor 3b, Subunit 1)

The *SF3B1* gene is responsible for encoding the splicing factor 3b subunit 1, and gene mutations occur in about 25% of MDS cases [42]. *SF3B1* mutations, known as the most common spliceosome lesions in MDS cases [2], have been reported to be associated with superior survival and a specific MDS subtype, namely MDS with ringed sideroblasts (MDS-RS) [43]. The study of Makishima et al. performed on a large group of MDS cases revealed that *SF3B1* gene mutations were mutually exclusive to both splicing factor mutations and recurrent gene mutations [44]. In addition, they found that *SF3B1* gene mutations were mutually exclusive with type-1 mutations (*PTPN11*, *FLT3*, *IDH1*, *WT1*, *NPM1*, *IDH2*, and *NRAS* gene mutations) and type-2 mutations (*TP53*, *GATA2*, *RUNX1*, *KRAS*, *STAG2*, *ZRSR2*, *ASXL1*, and *TET2* gene mutations) [44]. Moreover, it was observed that mutations in *JAK2* and *DNMT3A* genes significantly co-occurred with mutations that had an interest in the *SF3B1* gene [44].

In accordance with the fifth edition of the WHO Classification of haematolymphoid tumors, a distinct type of disease was proposed, namely MDS, with low blasts and *SF3B1* mutation (MDS-*SF3B1*) [1]. The prognostic impact of *SF3B1* gene mutation in MDS is favorable in cases with <5% bone marrow blasts and is neutral in cases with 5% to 30% blasts [45].

Considering that most of the MDS patients with *SF3B1* mutation presented a favorable clinical outcome and low risk of progression to leukemia [37,40,46], it may be assumed that *SF3B1* gene mutations are less frequent in cases with MDS transformation into AML than in MDS cases.

#### 3.1.2. SRSF2 (Serine/Arginine Rich Splicing Factor 2)

Mutations in the *SRSF2* gene were observed in 10–16% of MDS cases, which was associated with poor overall survival and had an adverse prognosis with an increased risk of transformation to AML [47,48]. Similarly, *SRSF2* mutations were shown to be independently associated with a negative prognosis impact for overall survival (hazard ratio HR = 2.3; 95% CI = 1.28–4.13; *p* = 0.017) and leukemic transformation (HR = 2.83; 95% CI = 1.31–6.12; *p* = 0.008) [49].

Spliceosome *SRSF2* mutations are the second most common alterations of the splicing factor in MDS [2] and are observed mainly in association with MDS characterized by multilineage dysplasia and are considered predictors of unfavorable prognosis and a high risk of AML transformation [40]. The presence of *SRSF2* mutation in MDS cases with <5% bone marrow blasts is associated with an adverse prognostic impact, while in cases with 5–30% blasts, it has a neutral impact [45]. 

The study of Wu et al. enrolled 223 Taiwanese MDS cases and detected *SRSF2* mutation in 34 (14.6%) of investigated patients and suggested that *SRSF2* mutation might have little impact on leukemic transformation [50]. The discrepancy results of Wu et al. may be explained by the fact that they did not analyze the prognostic impact according to the percentage of blasts. The same study found that *SRSF2* gene mutation was associated with the male gender, older age, CMML, and mutations of *ASXL1*, *RUNX1*, and *IDH2* genes, and it was stable during the progression of the disorder [50]. 

#### 3.1.3. U2AF1 (U2 Small Nuclear RNA Auxiliary Factor 1)

*U2AF1* gene mutations, observed in less than 10% of patients [36], have been described mainly in MDS patients characterized by multilineage dysplasia and excess blasts and showed inferior survival and an increased risk of leukemic evolution [37,40,46]. According to the available data, MDS cases with *U2AF1* gene mutations had a high probability of progression to AML (*p* = 0.03) [51]. *U2AF1* mutation had a negative impact on survival in MDS patients with blast percentages less than 5% but lost its significance in cases with 5–30% blast percentages [35,45]. Based on the data reported by Liu et al., the *U2AF1* gene was the most commonly mutated, and it was accompanied by trisomy 8 [35]. Wang et al. demonstrated that MDS cases with *U2AF1* and *ASXL1* gene mutations are prone to developing AML [52]. Additionally, the variant allele frequency (VAF) may provide prognostic information. In this respect, Wang et al. reported that MDS cases with a high mutation load (VAF  >  40%) of *U2AF1* had a short OS. In addition, they demonstrated that a high mutation load (VAF  > 40%) of *U2AF1* represented an independent factor of inferior survival [52].

#### 3.1.4. ZRSR2 (Zinc Finger (CCCG Type), RNA-Binding Motif and Serine/Argentine Rich 2)

*ZRSR2* gene mutations, found in around 3% of MDS patients [36], are more prevalent in MDS subtypes with no ring sideroblasts and CMML and are associated with an increased percentage of bone marrow blasts and an increased rate of transformation into AML. *ZRSR2* mutations are more common in MDS cases with *TET2* mutation [53]. The study of Jiang et al., which investigated the impact of VAF on the clinical outcomes of MDS, uncovered that higher a *ZRSR2* VAF was linked with shorter survival and suggested that VAF might represent an important factor for the prognostic implication of a specific gene [54].

### 3.2. Epigenetic Regulators

The mutations of genes that are implicated in post-translational modifications of histones and DNA (DNA methylation), which are important mechanisms of epigenetic regulation, are common in MDS [2]. DNA methylation interests the cytosine of cytosine-guanine (CpG) dinucleotides localized in the gene’s promoter and is considered to be associated with oncogenesis and leukemogenesis [5].

The mutations of genes that are involved in the epigenetic regulation of transcription (*DNMT3A*, *TET2*, *ASXL1*, *IDH1*, *IDH2*, and *EZH2*) are usually identified in MDS patients [40]. Previous studies suggested that epigenetic deregulation, for example, aberrant hypermethylation, may be involved in the silencing of tumor suppressor gene expression resulting in MDS progression to AML [55,56].

#### 3.2.1. ASXL1 (Additional Sex Comb-like 1)

*ASXL1* gene mutations, observed in 15–20% of MDS patients, are associated with an inferior prognosis in MDS cases, resulting in a shorter OS and an increased risk of progression to AML [17]. *ASXL1* mutation negatively affects OS in cases with blast percentages lower than 5% but loses its negative impact in cases with 5–30% blasts [45].

The study performed by Pellagatti et al. that investigated 41 MDS cases before and after the progression of MDS showed that *ASXL1*, encoding an essential epigenetic regulator, was the most frequently mutated gene, with a rate of mutation around 44–46% being strongly associated with leukemic transformation [57]. Tefferi et al. investigated 179 primary MDS patients that had a higher frequency (30%) of *ASXL1* mutation and observed that *ASXL1* mutations were less likely to co-exist with *SF3B1*, *U2AF1*, and *SRSF2* mutations [48]. In addition, they noticed that the mutations of the *ASXL1* gene were more likely to appear in cases with more mutations [48]. Liu et al. investigated 99 cases with MDS or MDS that progressed to AML and observed that *ASXL1* mutations occurred more frequently with *ETV6*, *RUNX1*, and *SRSF2* mutations [35].

#### 3.2.2. IDH1 and IDH2 [Isocitrate Dehydrogenase NADP(+)1 and Isocitrate Dehydrogenase NADP(+)2; Isocitrate Dehydrogenase Genes]

*IDH1* or *IDH2* genes could be observed in about 5% of MDS cases [31] and were involved in the production of enzymes that are involved in the process that converts isocitrate to 2-ketoglutarate to generate cellular energy [42].

The study of Lin et al. indicated that mutations in *IDH1* did not significantly increase the risk of transformation into acute leukemia, while the presence of *IDH2* mutation in MDS cases was associated with an increased risk of evolution to AML (*p* = 0.004) [58]. Moreover, the same study observed that the mutation of the *IDH2* gene represented an independent predictor of poor survival (*p* = 0.04) and a shorter duration of leukemia-free survival (*p* = 0.04) [58]. These findings are in line with that of Jin et al. [59] but are contradictory to the previous study performed by Patnaik et al. [60]. *IDH1* and *IDH2* mutations are targets for new treatments and are associated with MDS and excess blasts and MDS with multilineage dysplasia [45,61]. *IDH2* mutations had a higher prevalence compared with *IDH1* mutations and were associated with *ASXL1*, *DNMT3A*, and *SRSF2* gene mutations [36]. A cooperative relationship (correlation coefficient < 0.001) was noticed between *IDH2* and *DNMT3A* mutations [62]. Such a cooperative relationship may explain the conflicting results of the mentioned studies.

#### 3.2.3. TET2 (Ten-Eleven Translocation Proteins)

*TET2* gene mutations were identified in 4–12% of MDS patients and did not coexist with *IDH1*, *IDH2*, or *RUNX1* mutations [62]. Nazha et al. reported that *TET2* mutations were commonly associated with normal cytogenetic analysis or normal karyotype, and its occurrence with *SRSF2* or *ZRSR2* gene mutations had been established as predictive for the transformation to AML and characteristic for CMML [45]. 

MDS cases with *TET2* mutation had a shorter period of time for progression to AML (HR = 7.81; 95% CI: 2.08–29.31) and tended to show an inferior survival in cases included in the very high-risk group of IPSS-R (HR = 2.02; 95% CI = 0.77–5.36) [63].

The study of Jiang et al. also reported that *TET2* VAF played an important role in leukemic transformation [54]. It was demonstrated that *TET2* VAF was independently associated with faster leukemic progression (hazard ratio HR = 1.013 per each 1% VAF increase; 95% CI = 1.005–1.022; *p* < 0.05) [54]. Considering that MDS is a hematological malignancy that presents clonal hematopoiesis, it is important to establish the VAF of certain genes, and Jiang et al. recommended a routine investigation of the mutational VAF of certain genes (TET2, TP53, ZRSR2, *RUNX1*, and *DNMT3A*) for the outcome prediction and leukemic transformation [54].

#### 3.2.4. DNMT3A (DNA Methyltransferase 3 Alpha)

The *DNMT3A* gene is implicated in the methylation of DNA and promotes the differentiation of HSC into progenitor cells [42]. *DNMT3A* mutations were identified in about 10% of MDS cases and were found to be associated with MDS with multilineage dysplasia, MDS with excess blasts [45], and inferior prognosis [20]. A recent study suggested that *DNMT3A* gene mutations tend to be associated with transformation into leukemia (HR = 1.516; *p* = 0.098) [54], and it showed that increased *DNMT3A* VAF was associated with poor survival [54] underlying the predictive usefulness of *DNMT3A* mutational VAF for the prognostic assessment of MDS. It was shown that mutations in epigenetic modifiers, particularly in *DNMT3A* and *TET2* genes, tend to appear early in the evolution of MDS [2]. Patients with *DNMT3A* mutation had a higher risk for progression to leukemia, with an overall hazard ratio of 6.87 (*p* < 0.05, 95% CI = 2.80–16.87) [64].

#### 3.2.5. EZH2 (Enhancer of Zeste Homolog 2, Enhancer of Zeste 2 Polycomb Repressive Complex 2 Subunit)

*EZH2* mutations, an important member of Polycomb group complex 2 (PRC2), were found in 6% of MDS cases and co-occurred with *RUNX1* gene mutations [36]. *EZH2* gene mutations are described as associated with oncogenesis as well as with the progression of cancers [65] and have been reported to confer an unfavorable impact on overall survival being associated with leukemic progression (HR = 2.536; *p* = 0.002) [54]. Recently, it was demonstrated that MDS transformation to AML was independently associated with the mutation status of *EZH2* [54]. 

### 3.3. Transcription Factor Genes

Part of the gene mutations that appeared during MDS transformation into AML is represented by those in core hematopoietic transcription factor genes, which include the following genes *RUNX1*, *GATA2*, and *CEBPA* (CCAAT-enhancer binding protein α), and which interfere with the normal process of differentiation [35,66]. 

#### 3.3.1. RUNX1 (Runt-Related Transcription Factor 1)

*RUNX1*, a critical transcription factor gene, has consistently been associated with an adverse prognosis in MDS cases [9]. *RUNX1* mutations were identified in 5–9% of MDS cases and were associated with higher marrow blast percentage and with *SRSF2* gene mutations [36].

MDS and its progression to AML are linked with the acquisition of different gene mutations, particularly the *RUNX1* gene mutation [17]. Similarly, the meta-analysis of Sutandyo et al. showed that adult MDS cases with *RUNX1* mutations were associated with MDS transformation into AML (HR = 1.85; 95% CI = 1.11–3.09; *p* = 0.02) [64].

The study performed by Cho et al. revealed that *RUNX1* was the most frequently mutated gene in MDS transformation to AML and suggested that mutations that interested the *RUNX1* gene may appear later in the process of tumorigenesis and may be associated with a poor prognosis due to evolution to a more aggressive disorder [39].

#### 3.3.2. GATA2 (GATA Binding Protein 2)

*GATA2* mutations were reported in about 14% of MDS patients that progressed to AML [30]. The data regarding the impact of *GATA2* mutations on survival were contradictory. Xu Y et al. reported an inferior OS (HR = 3.71) [67], while no influence on the OS was found in another study (HR = 1.19, 95% CI = 0.53–2.66) [12]. Meggendorfer et al. concluded that mutations in *GATA2*, *ASXL1*, *IDH2*, *RUNX1*, *NRAS*, *SRSF2*, and *ETV6* genes might predispose a transformation to leukemia [23]. It was observed that MDS cases with *GATA2* mutation presented cytogenetic aberrations: the most frequent were chromosome 7 anomalies such as monosomy 7 and der (7) in 41% of investigated patients, followed by the gain of chromosome 8 (trisomy 8) in 15% of patients, while complex karyotype and deletion 5q were very rare or absent [68].

#### 3.3.3. CEBPA (CCAAT Enhancer Binding Protein Alpha, α)

Most MDS patients have a single *CEBPA* mutation, but both mutations were observed in AML cases secondary to MDS. The presence of a single *CEBPA* mutation coupled with mutations in different genes showed a poor prognosis in MDS [69,70,71]. The study of Shih revealed that the frequency of *CEBPA* gene mutations was 8% at the moment of MDS diagnosis and 12% at the progression from MDS to AML [69] and considered that CEBPA mutations might be involved in the pathogenesis of a subgroup of MDS cases with the progression of the disease [69]. *CEBPA* mutations are more common in secondary AML, indicating that these gene mutations are attained later during the progression of the disease to AML in a subgroup of cells that extend [17,70].

#### 3.3.4. BCOR and BCORL1 (Components of a Polycomb Repressive Complex PRC)

Mutations in *BCOR* genes were observed in approximately 5% of patients with MDS, and frameshift mutations were associated with an unfavorable outcome with inferior OS (HR = 3.3) [37,42]. *BCORL1* mutations were associated with MDS progression to leukemia, and *BCORL1* VAF (HR = 1.025, *p* = 0.081) tended to be linked to the leukemic progression of MDS [54]. Moreover, it was observed that *BCOR* gene mutations were frequently associated with *DNMT3A* and *RUNX1* mutations [42].

#### 3.3.5. ETV6 (Ets Variant 6)

*ETV6* is responsible for encoding a transcriptional repressor, and *ETV6* mutations or dysregulated gene expression can lead to the development of leukemia or leukemogenesis [72]. *ETV6* mutations are rare in MDS (3%) and correlate with shorter survival and a variable predisposition to leukemia [36]. 

### 3.4. Cell-Cycle Regulators

#### TP53

The *TP53* gene localized on chromosome 17 is a tumor suppressor gene and a transcription factor that induces apoptosis, the arrest of the cell cycle, and allows DNA repair to protect cells against stress and damage [73]. The presence of *TP53* gene variants in MDS is associated with high-risk disease progression with rapid transformation to AML, independently of the revised International Prognostic Scoring System (IPSS-R), resistance to treatment, and dismal outcomes [10,74,75].

*TP53* gene mutations were detected in 8–13% of MDS patients, were associated with complex karyotypes [36], and were mutually exclusive with RNA splicing factor gene mutations, mainly with the *U2AF1* gene mutation [35].

It was observed that subclones representing *TP53* gene mutations might occur at an early stage of the disorder in MDS cases with del(5q) and predict a poor response to lenalidomide [35]. A higher risk related to the leukemic transformation and shorter overall and event-free survival was noticed amongst cases with the mutation of the *TP53* gene and isolated 5q deletion (del 5q) treated with lenalidomide [76]. The *TP53* mutation status should be considered for diagnosis, prognostic, and before therapy decisions [10,77]. Recently, it was suggested that biallelic *TP53* lesions are a potent driver of MDS progression, which reinforced the significance of investigation into the *TP53* allelic state both for diagnosis, monitoring the disease, and the identification of high-risk MDS cases [10].

The study performed by Meggendorfer et al. showed that *TP53* mutations were more often identified in MDS cases with isolated del(5q) in comparison to all other MDS subtypes [77]. The leukemic progression of MDS cases was driven by the unfavorable prognostic impact of *TP53* gene mutation in association with the deletion of chromosome 5 [del(5q)] and potentially the acquisition of *RUNX1* gene mutation [77]. 

A recent meta-analysis that included 4003 MDS cases and 1278 patients with *TP53* gene mutations investigated the impact of the VAF of the *TP53* mutation and showed that a high VAF was a severe and independent prognostic factor for survival in MDS cases with *TP53* mutation [78]. The predictive power of mutational VAF in MDS progression to AML was evaluated in several studies. Jiang et al. communicated that *TP53* VAF was associated with a faster leukemic transformation [54]. The impact of the clonal burden of *TP53*, *DNMT3A*, *TET2*, and *NPM1* mutations on survival time was investigated, and it was observed that an increased VAF indicated a lower OS and a high risk for leukemic transformation [54]. Those results are, as expected, being taken into consideration due to their association with a poor prognosis in patients. VAF is important for every somatic mutation, not only in the case of MDS patients.

### 3.5. Cell-Signaling Molecules

Mutations in signaling pathway components occur during MDS transformation, such as *FLT3* (fms-related tyrosine kinase 3) and *RAS* family members that are involved in cell proliferation control. The presence of these mutations in low-risk MDS was reported to be associated with progression to AML [2].

#### 3.5.1. RAS Pathway

*RAS* mutations are considered to be important genetic events involved in the pathogenesis of acute leukemia; therefore, the analysis of these mutations during the course of MDS is potentially useful as an indicator of leukemic progression [23]. *RAS* mutations occur later in disease evolution and are mostly subclonal events. 

Badar et al. reported that patients with *RAS* mutation had a median survival time after MDS progression to leukemia of 3.6 months, whereas the survivability increased to 7 months in cases without *RAS* mutation (hazard ratio HR = 2.44, 95% CI = 1.80–6.72, *p* = 0.0008) [24].

The presence of *RAS* mutations not only in high-risk MDS cases but also in low-risk ones was associated with impending transformation, indicating that even low-risk cases have small subclonal populations that are predictive of leukemic transformation and a reduced OS. 

According to the findings of Park et al., *KRAS* mutations confer significantly inferior survival rates compared with mutations in other genes (median OS was 0.4 vs. 6.5 months, *p* = 0.007) [79]. Moreover, *NRAS* mutations co-occur with other gene mutations, such as *KRAS* and *CEBPA* (*p* = 0.012; and *p* = 0.049, respectively) [79]. The study performed by Badar et al. indicated that the acquisition of detectable levels of *RAS* and/or *FLT3*-ITD gene mutation at the moment of MDS transformation to AML resulted in approximately 30% of cases and predicted extremely poor outcomes [24]. Shih et al. established this regarding 33% of patients with MDS acquiring the *FLT3* or *NRAS* gene mutations during progression to AML [80], and once the patients with these anomalies developed secondary AML, the prognostic was directly correlated with the VAF. Generally, *FLT3* and *RAS* mutations are associated with poor OS.

*NRAS* and also *ASXL1*, *RUNX1*, and *SETBP1* gene mutations were proved to be independent risk factors for inferior OS and the increased risk of MDS progression to leukemia [81].

#### 3.5.2. PTPN11 (Protein Tyrosine Phosphatase Non-Receptor Type 11)

*PTPN11* gene mutations are uncommon in MDS [42], identified in 2.8% of patients [11], and the data regarding their impact on outcomes are conflicting [37,44]. A higher frequency of *PTPN11* mutations in MDS cases with progression to AML (17.86%) compared with cases with MDS (2.82%) was observed [35]. No survival impact was found for *PTPN11* mutations in previous studies [37,82], but according to the study of Makishima H et al., these gene mutations are associated with faster disease progression to AML and a lower overall survival time [44]. Similarly, Wu et al. found that *PTPN11* gene mutations correlated with survival (HR = 2.03; 95% CI = 1.12–3.69; *p* = 0.02) [11].

#### 3.5.3. CBL (Casitas B-Cell lymphoma)

*CBL* has a major role in tyrosine kinase signaling, and it is also involved in the degradation of some important proteins (FLT3, c-Kit, and STAT5) in myeloid neoplasms [83]. *CBL* mutations occur less frequently in MDS, in about 1.8% of cases [11], and are considered to be late events. There is evidence regarding the negative impact of gene mutations with survival being associated with an adverse prognosis [42,84,85]. Dan et al. suggested that *CBL* gene mutations were associated with more aggressive types of MDS and that there were implicated in disease progression to AML [5].

#### 3.5.4. FLT3 (FMS-like Tyrosine Kinase Gene 3)

*FLT3* mutations are rare events in MDS but are among the most common mutations in AML. 

Meggendorfer M et al. evaluated 38 patients who were investigated at the moment of diagnosis of MDS and later at their progression to AML and observed an *FLT3* mutation only in cases with leukemic transformation (16% with *FLT3*-ITD and 8% *FLT3*-TDK mutations, respectively) [23]. Similar results were observed by Badar et al.; in 102 MDS cases [24], *FLT3*-ITD mutations were identified in 19% of patients at the moment of transformation to AML. The median survival after leukemic transformation in cases that harbored *FLT3*-ITD mutations was 1 month compared to 6 months in patients without *FLT3*-ITD mutation (hazard ratio HR = 3.08, 95% CI = 2.1–15.76, *p* < 0.0001) [24]. The study performed by Shih et al. revealed an association between *FLT3*-ITD and an adverse outcome due to faster progression to AML and shorter survival [69]. 

Therefore, the acquisition of *FLT3* mutation drives the MDS progression into AML and should be considered a marker of disease progression. 

Furthermore, the study conducted by Badar et al. demonstrated that the acquisition of *FLT3*-ITD and/or *RAS (NRAS*, *KRAS)* mutations at the moment of MDS transformation into AML was found in 26% of cases and was associated with extremely poor outcomes with a median survival of 2.4 months [24].

Takahashi et al. analyzed the incidence of the dynamic acquisition of *FLT3* and *RAS* gene mutations in low-risk MDS and its effect on transformation to AML and survival [86]. Takahashi et al. observed the acquisition of *FLT3* or *RAS* mutations in 23% of the cases from 74 with leukemic transformation [86], suggesting their role in driving leukemic transformation. Moreover, it was observed that the acquisition of *FLT3* and *RAS* mutations were almost mutually exclusive [86]. Takahashi et al. documented the transformation to leukemia in 90% of patients with *FLT3* or *RAS* mutation acquisition with a median time to the transformation of about 11 months [86]. By multivariate analysis, a very strong correlation between *FLT3* (*p* = 0.004) or *RAS* (*p* = 0.002) mutation acquisition and worse survival was reported [86]. 

A series of studies revealed that the presence of *NRAS*, *FLT3*, or *PTPN11* gene mutations was associated with faster MDS transformation into AML [44,57,87]. According to the findings of Makishima et al., type-1 mutations (*FLT3*, *PTPN11*, *NRAS NPM1*, *IDH1*, *IDH2*, and *WT1*, gene mutations) were acquired during MDS transformation to AML [44]. The presence of type-1 mutations in MDS cases was considered to be associated with the increased risk of evolution to leukemia and shorter overall survival compared to the presence of other mutations [44]. In addition, MDS cases with type-1 mutations showed a significantly faster progression to leukemia than those without type-I mutations [29].

Therefore, a close follow-up of MDS cases that achieved type-1 mutations might allow for an early diagnosis of MDS leukemic evolution. In addition, it is important to use the latest high-resolution technologies, such as NGS, for the early detection of small subclones and, therefore, for the early detection of the risk of MDS progression, considering that this type-1 mutation usually precedes transformation to AML.

Based on the study performed by Bernard et al., strong predictors of adverse outcomes in MDS were represented by *TP53*, *FLT3*, and *MLL* gene mutations [88].

#### 3.5.5. JAK2 (Janus Kinase 2)

It has been reported that approximately 5% of MDS cases with the isolated loss of 5q or deletion of the long arm of chromosome 5 also harbor the *JAK2* V617F gene mutation [89,90]. Different studies reported MDS transformation to AML with a frequency between 6% [72,82] and 12.8% [91]; one possible explanation may be the simultaneous presence of deletion 5q and *JAK2* V617F gene mutation and other gene mutations (for example, *TP53*). It was observed that MDS cases with concomitant 5q deletion and *JAK2* V617F mutations were associated with ACA during disease progression to leukemia [90,92].

Sangiorgio et al. reported no statistical differences regarding the MDS progression to AML or overall survival between the MDS del(5q) with and without *JAK2* mutation [91]. The impact of *JAK2* V617F gene mutation on survival in MDS cases was unclear.

### 3.6. Other Genes

#### Nucleophosmin Gene (*NPM1*)

*NPM1* mutations were identified in about 2–3% of MDS cases [15,93]. MDS patients with *NPM1* mutation had a poor clinical course and were more likely to progress into AML [15,94]. The negative impact of *NPM1* mutation on overall survival was observed in more studies that included 508 MDS cases and 944 MDS patients, respectively [37,45]. MDS with *NPM1* mutation should be regarded as an early-stage AML rather than MDS [95].

### 3.7. Relation between Chromosomal Abnormalities and Gene Mutation

Trisomy 8 frequently coexisted with the *U2AF1* gene mutation [35,62] and also with *ZRSR2* mutations [62]. Generally, trisomy 8 has a neutral impact but is associated with several genetic anomalies. By this, the patient’s prognostic is influenced by the additional mutation. Therefore, patients with trisomy 8 should be thoroughly investigated. Moreover, the study of Xu et al. found that the loss of chromosome 20, del(20q) coexisted with *SRSF2*, *U2AF1*, and *WT1* mutations [62]. Chromosome 7 abnormalities [−7, del(7q)] often coexisted with *SETBP1* and *RUNX1* gene mutation and were associated with poor outcomes [62]. Additionally, del(5q) co-occurred frequently with *SF3B1* and *TP53* mutation [62]. *U2AF1*, *RUNX1*, or *TP53* gene mutations were less probable to co-exist with normal karyotype, and a strong correlation was observed between *TP53* gene mutation and complex karyotype [62,79]. 

The genetic landscape of MDS and the progression of MDS to leukemia is complex due to chromosomal abnormalities and somatic mutations. The accumulation of epigenetic mutations represents a significant factor in AML transformation. *TET2* and *IDH1/2* gene mutations are driver mutations obtained during MDS’s evolution to AML. In addition, type-1 gene mutations (*FLT3*, *PTPN11*, *NRAS NPM1*, *IDH1*, *IDH2*, and *WT1*) were acquired during MDS progression to AML and are considered to be associated with faster leukemic transformation. Co-mutation of *TET2* and *SRSF2* genes represents an important marker for the leukemic transformation of MDS. It was observed that the presence of particular gene mutations and the co-occurrence of certain gene mutations are predictive for leukemic transformation (Table 2). Therefore, the presence of mutations that interest the signaling genes (such as *NRAS*, *KRAS*, *PTPN11*, *CBL*, *JAK2*, *FLT3*) and IDH1 and IDH2 mutations could lead to the MDS’s progression into AML. Moreover, identifying the presence of the above-mentioned gene mutations could be useful in treatment decision-making.

Recent advances in genomics have revealed that particular gene mutations and the co-occurrence of certain gene mutations are essential predictors of prognosis and leukemic transformation. Correct diagnosis and reliable risk stratification are important also for those with disease progression to acute leukemia that may benefit from targeted therapy and novel therapies for improving overall survival. 

The present review presents the cytogenetic aberrations observed in MDS, and during disease progression to AML, the molecular profile focused on somatic mutations and also the relation between chromosomal abnormalities and gene mutations. This work provides a comprehensive review of the genetic anomalies implicated in the pathogenesis, diagnosis, risk stratification, and latest targeted therapeutic approaches for MDS and in MDS progression to AML. The review is focused on the most recent scoring systems that allow for an improved prediction of survival, prognosis, and the risk of progression to leukemia and for proper therapeutic decisions. 

The limitation of the present work is represented by the lack of comprehensive discussion of germline mutation. It is known that there are heterogeneous conditions that may associate with subtle or mild symptoms and that are associated with the predisposition to a myeloid neoplasm that may progress to MDS. In addition, the MDS classification that was available did not include specific subtypes of MDS associated with mutations in the splicing genes (for example, *U2AF1* and *SRSF2* genes which are associated with an unfavorable prognosis) or in epigenetic modifiers (such as *ASXL1*, *TET2*, *DNMT3A*, etc.) were associated with poor outcomes. 

## 4. Treatment of Myelodysplastic Neoplasm

### 4.1. Current Treatment of Myelodysplastic Neoplasm

Due to the genetic heterogeneity of MDS, the therapeutic options currently available for MDS are limited. The treatment of MDS is influenced by the disease characteristics, age of the patient, and comorbidities which vary from supportive care to hematopoietic cell transplantation [96]. As can be observed, current therapeutic options in cases diagnosed with MDS are recommended in accordance with the patient’s risk stratification, which is based on scoring systems (IPSS and IPSS-R). In patients with low-risk MDS, the aims of the treatment are represented by the improvement of cytopenias, especially the symptoms related to anemia, and the reduction in the number of required transfusions [96,97]. In high-risk MDS patients, the aim of the treatment is to prevent the progression of the disease and to improve survival [98].

The most important concern in the management of MDS cases is represented by the risk of MDS progression to AML, which may be prevented by using drugs that interfere with the disorder’s natural history.

Erythropoiesis stimulating agents (ESA) such as recombinant humanized erythropoietin or the longer-acting erythropoietin, darbepoetin alfa, is considered the standard first-line therapy for anemia in low-risk MDS patients [3,97]. In low-risk MDS cases with del(5q) with or without additional cytogenetic aberrations, the recommended treatment is represented by Lenalidomide (LEN) which resulted in erythroid responses in about 70% of the cases [3,43]. In low-risk MDS cases with an inferior response to monotherapy with Lenalidomide or those cases that were refractory to the ESA, a combination therapy of Lenalidomide and ESA was proposed to achieve an erythroid response in about 40% [99] and transfusion independence in about 20% [3]. The combination of Lenalidomide and ESA did not influence the response duration [99]. 

Hypomethylating agents (HMA) represent a common treatment approach for MDS patients [3]. HMAs, such as azacitidine or decitabine, are recommended for patients with high-risk MDS [97] HMA increased survival in MDS cases, improved the quality of life, and the outcomes in high-risk MDS, and were considered the standard of care treatment in high-risk MDS patients until disease progression or intolerance [3,98]. Azacitidine and decitabine are approved for high-risk MDS patients who are not eligible for intensive chemotherapy, but only 50–60% of patients respond to this treatment, and the duration of the response is usually less than 2 years [3,98].

In high-risk MDS intensive chemotherapy, an anthracycline-cytarabine combination or high-dose cytarabine may be considered. Intensive chemotherapy may induce complete remission in high-risk MDS cases [98]. Volpe et al. recommended intensive chemotherapy in young high-risk MDS cases for those who needed significant cytoreduction and were eligible for hematopoietic cell transplantation [3]. 

The most efficient preventive treatment for MDS progression to AML is considered allogeneic stem cell transplantation (allo-SCT) [97], and it should be considered as soon as possible for high-risk MDS patients at the time of diagnosis. For MDS cases who are not eligible for SCT, hypomethylation therapy should be started, and it should continue until MDS progression. It was suggested that new treatment options, such as hypomethylating agents (HMA) or possibly lenalidomide, could decrease the risk of MDS transformation into leukemia [55]. 

### 4.2. Novel Therapies in Myelodysplastic Neoplasm 12%–13

The mutational status of a patient with MDS is related to a specific subtype and outcome. These cases should be considered for medication responsiveness. 

Encouraging results have been obtained (Table 3), for example, for the use of *IDH* inhibitors (enasidenib and ivosidenib) for MDS cases with *IDH1* and *IDH2* gene mutations or lenalidomide for those with del(5q). MDS cases with *SF3B1* mutations may benefit from new drugs (namely, Luspatercept, a recombinant fusion protein). The product of the *SF3B1* gene binds to transforming growth factor β (TGFβ) superfamily ligands to decrease *SMAD2* and *SMAD3* signaling, thus allowing erythropoiesis (erythroid maturation) [43,100]. Intensive chemotherapy and allo-SCT seem to improve the outcome of MDS patients with *NPM1* mutation [101]. Unfortunately, only about half (50%) of MDS cases respond to HMA: the present standard of care in high-risk MDS. Nevertheless, most of the responding MDS cases eventually progress. Venetoclax is a novel orally selective inhibitor of the anti-apoptotic protein BCL-2. Venetoclax and HMA induce high response rates in MDS, including relapsed/refractory MDS cases [102].

Recently, it was reported that the combination of HMA (azacitidine) with venetoclax in high-risk MDS might lead to a higher response rate. The combination of HMA and venetoclax was approved for high-risk MDS patients ineligible for allo-SCT [106]. 

The new generation of hypomethylating agents is in clinical trials in high-risk MDS cases. Guadecitabine, a novel HMA, is recommended in high-risk MDS in first-line therapy with good response (about 60%) [98].

CPX-351, a combination of cytarabine and daunorubicin, approved for the treatment of AML-MR, was reported to be effective in high-risk MDS patients, especially in attaining marrow blast clearance and as a bridge to allo-SCT [107].

Altogether, these data show that the identification of MDS at risk of transformation into AML is critical as new treatment approaches are available and may improve the outcome. Regular clinical follow-up of MDS patients and evaluation of factors that show disease progression in high-risk patients are recommended, aiming to implement early specific treatment. 

## 5. Conclusions

The advanced research has allowed the discovery of molecular features for the leukemic progression of myelodysplastic syndromes, clarifying their impact on disease and prognosis and revealing novel diagnostic and prognostic markers. *FLT3*, *IDH1* and *IDH2*, *TP53* genes, co-mutation of *TET2* and *SRSF2* genes, increased VAF for *TP53*, *DNMT3A*, *TET2*, and *NPM1* genes mutations, and the presence of ACA may be considered important markers for the leukemic transformation of MDS. The uncovering of new markers of MDS predicting leukemic progression and patient survival will improve patient stratification, resulting in more tailored and efficient therapeutic approaches. 

There are certain patterns in the combinations of genetic abnormalities in MDS patients and MDS progression to AML, which may be useful for precision prognostication, precision treatment, prediction of response to the therapy, and predicting of progression to AML. 

Unraveling molecular driver anomalies is crucial to identify patients who are at high risk for leukemic progression, as these should benefit from personalized treatment and should be considered as soon as possible for hematopoietic cell transplantation. Moreover, the identification of the molecular landscape of MDS is useful for treatment decision-making and direct novel treatments. Considering that MDS is characterized by a high risk of transformation into leukemia that appears in approximately 30–40% of MDS patients and that there are associated with specific MDS subtypes, therapy responses, and clinical outcomes, it is crucial to know the genetic anomalies that help clinicians for better clinical management.

Even if genetic anomalies of MDS are known, our manuscript highlighted cytogenetic and molecular anomalies that may appear during disease evolution (RNA-splicing factors, epigenetic regulators, transcription factors, cell-cycle regulators, cell-signaling molecules, as well as other gene mutations), which are important for the prediction of OS and transformation to AML, more studies are needed in order to identify new potential markers of leukemic progression in MDS patients. The implementation of genomic methods in clinical practice may contribute to a more refined detection of genetic anomalies and therefore allow a faster diagnosis of disease progression to leukaemia in earlier stages.

## Figures and Tables

**Figure 1 ijms-24-05734-f001:**
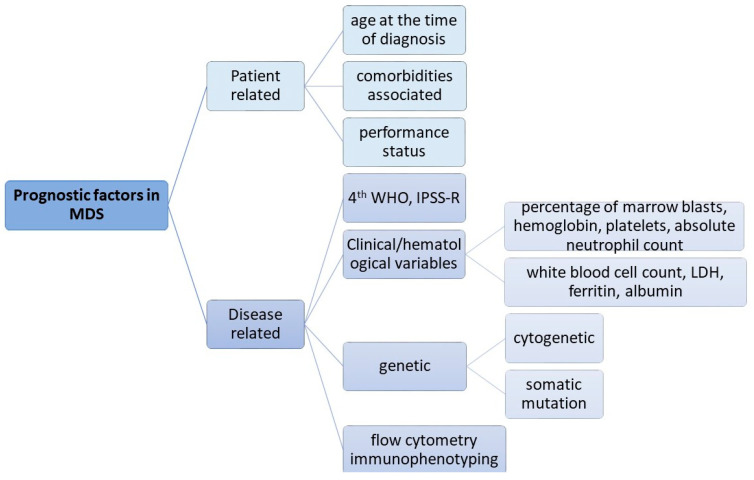
Prognostic factors in MDS (WHO—World Health Organization classification; IPSS-R—Revised International Prognostic Scoring System; LDH—lactate dehydrogenase).

**Figure 2 ijms-24-05734-f002:**
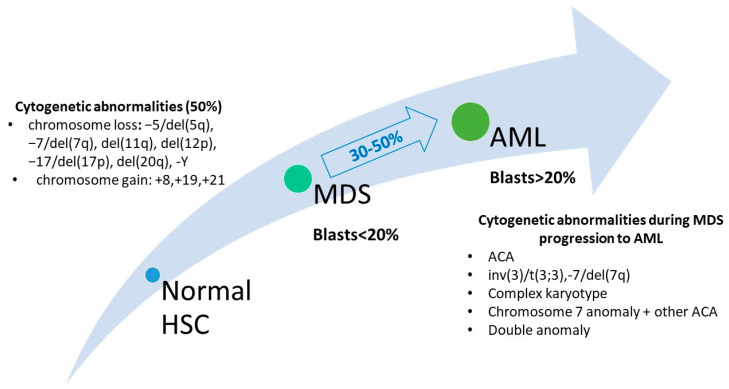
Chromosomal aberrations in MDS and in MDS progression to AML (HSC—hematopoietic stem cell; ACA—additional chromosomal abnormalities; del—deletion; inv—Inversion, t—translocation).

**Figure 3 ijms-24-05734-f003:**
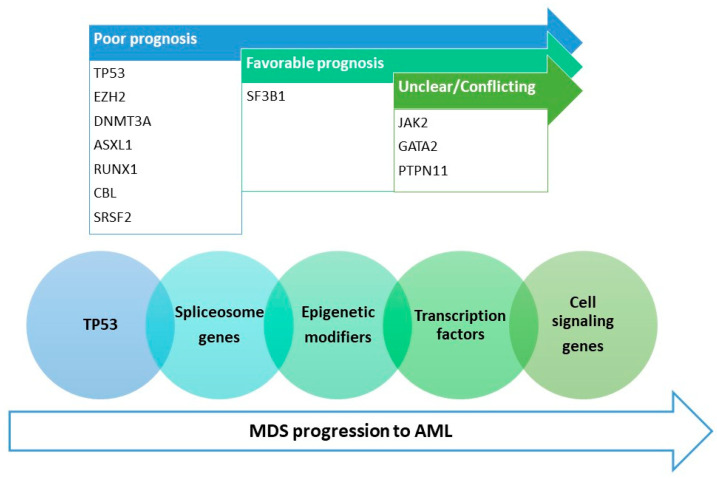
Gene mutations implicated in MDS progression to AML according to prognostics and the order of mutation acquisition (MDS—myelodysplastic syndrome, AML—acute myeloid leukemia, the order of acquisition of the mutations during MDS evolution to AML is represented with blue arrow).

**Table 1 ijms-24-05734-t001:** Characteristics of different prognostic scoring systems in MDS.

Score System	IPSS [7]	IPSS-R [8]	IPSS-M [10]	PPM-MDS [12]
Year of description	1997	2012	2022	2021
Type	clinical-cytogenetic model	clinical-cytogenetic model	clinical-molecular prognostic model	apply machine-learning techniques to clinical-molecular data
Risk subgroups	4 risk categories	5 risk categories	6 risk categories	similar to IPSS-R
Variable included	hematologic parameters, cytogenetic abnormalities; presence of cytopenia and clinical data	hematologic parameters, cytogenetic abnormalities, clinical data	hematologic parameters, cytogenetic abnormalities, somatic mutations	clinical, pathological, and genomic data
Advantages	used for evaluating prognosis in untreated adult MDS cases, at diagnosis.	provide better risk stratification and an accurate prediction of OS and transformation to AML; validated for treated and untreated MDS.	provide a personalized risk score and improved prognostic discrimination across all clinical endpoints, survival, and AML transformation; applicable for treated and untreated MDS.	outperformed IPSS-R; improved prediction of OS and transformation to AML; allow a more proper risk group
Disadvantages/limitations	not a precise predictor of prognosis in low-risk MDS; attributes little weight to cytogenetics;	does not incorporate mutational data; included patients who would now be diagnosed as AML (marrow blasts ≥20).	high performant infrastructure needed	the impact of uncommon mutations on outcomes can be too low and misleading.

PPM-MDS—Personalized Prediction Model for MDS; OS—overall survival, IPSS—International Prognostic Scoring System, IPSS-R—revised International Prognostic Scoring System, IPSS-M—molecular International Prognostic Scoring System, AML—acute myeloid leukemia.

**Table 2 ijms-24-05734-t002:** Gene mutation in MDS and in progression to leukemia.

Gene	Gene Function/Role in Cancer	Prognosis	Other Genetic Anomalies That Co-Occur
*SF3B1*	oncogene	favorable prognostic impact in MDS with <5% blasts neutral impact in cases with 5–30% blasts	
*SRSF2*	oncogene	adverse	*RUNX1*, *IDH2*, *ASXL1*, *TET2*
*U2AF1*	oncogene	adverse	trisomy 8, complex karyotype
*ZRSR2*	tumor suppressor	adverse	trisomy 8
*ASXL1*	tumor suppressor	adverse	*ETV6*, *RUNX1*, and *SRSF2* mutation
*IDH1*	oncogene	neutral impact	
*IDH2*	oncogene	adverse	*DNMT3A*, *ASXL1*, *SRSF2* mutations and also with *NPM1*
*TET2*	tumor suppressor	adverse	*SRSF2*
*DNMT3A*	tumor suppressor	adverse	
*EZH2*	tumor suppressor	adverse	*RUNX1*, *TET2*
*RUNX1*	tumor suppressor/oncogene	adverse	*SRSF2* mutation, complex karyotype
*GATA2*	tumor suppressor/oncogene	conflicting (adverse or neutral impact on OS)	chromosome 7 anomalies, trisomy 8
*CEBPA*	tumor suppressor/oncogene	adverse	
*BCOR*	tumor suppressor	adverse	*RUNX1* and *DNMT3A* mutations
*ETV6*	tumor suppressor	adverse	
*TP53*	tumor suppressor/oncogene	adverse	complex karyotypes
*RAS (KRAS*, *NRAS)*	oncogene	adverse	*CEBPA*
*PTPN11*	oncogene/tumor suppression	conflicting (adverse or neutral impact on OS)	
*CBL*	oncogene	adverse	
*FLT3*	oncogene	adverse	
*JAk2*	oncogene	unclear	
*NPM1*	oncogene/tumor suppressor	adverse	

OS—overall survival.

**Table 3 ijms-24-05734-t003:** Selected therapies for MDS [9,98,103,104,105].

Treatment	Indication	Notes
allo-SCT	Patients previously exposed to multiple therapies (growth factors, lenalidomide, HMA), should be considered for transplantation, and patients who fail to lenalidomide or azanucleoside.	Not recommended for cases with low-risk MDS; Due to age and comorbidities, most MDS cases are not eligible for allo-SCT, even if it represents the only curative option; MDS patients with del(5q) who harbor/develop *TP53* mutation during lenalidomide treatment should be considered for human cell transplantation.
Lenalidomide (LEN)	MDS cases with a deletion of chromosome 5, del(5q)	This may lead to longer survival for patients that respond to therapy; *U2AF1* gene mutations may be associated with a reduced probability of response in MDS cases.
Hypomethylating agents (HMA)	Low-risk MDS, High-risk MDS	HMAs are commonly used for the treatment of MDS; Azacitidine has significant clinical benefits in high-risk MDS patients; HMA represents the standard of care treatment in high-risk MDS cases (that are not candidates for allo-SCT or until disease progression or intolerance); HMA treatment may be recommended for low-risk MDS that are refractory to first-line treatment with growth factors, LEN, and/or luspatercept (ACE-536).
Azanucleosides (Azacitidine, Decitabine, ASTX727)	High-risk MDS	They are considered the standard of care for almost all cases with high-risk MDS; ASTX727 is an oral HMA, that consists of a combination of decitabine and cedazuridine, and was authorized for high-risk MDS.
Immunosuppressive Therapy (IST)	Low-risk MDS	MDS cases who failed treatment with HMA; Not supported by older MDS cases.
Luspatercept	MDS cases with failure of response to HMA; Low-risk MDS cases with failure to ESA, or are intolerant to ESA	Is a transforming-growth factor beta (TGF-β) ligand that induces the downregulation of the pathogenic SMAD2/SMAD3 in cases with ineffective erythropoiesis states; Low-risk MDS with ring sideroblasts and/or *SF3B1* mutation
Sotatercept (ACE-011)	Low-risk MDS cases with failure to ESA	Is a TGF-β inhibitor, a new activin-receptor fusion protein,
Venetoclax	MDS patients with failure of response to HMA	It is a highly inhibitor of the antiapoptotic molecule, inhibitor of BCL2 (B-cell leukemia/lymphoma-2); A phase I study for the treatment of cases with relapsed or refractory high-risk MDS noticed the potential benefit of the addition of venetoclax to HMA with an 87% overall response and higher OS; Despite the addition of Venetoclax, the presence of *TP53* gene mutations and complex karyotypes was associated with inferior prognosis.
Imetelstat	Low-risk MDS patients with adverse outcomes	Novel telomerase inhibitor; It is indicated for the refractory group of low-risk MDS cases with unfavorable outcomes (heavily transfused cases with-low risk MDS who are ineligible for or relapsed or refractory to erythroid stimulating agents (ESA), recombinant erythropoietin and darbepoetin; ESAs are usually the first-line agents recommended for the treatment of anemia in low-risk MDS cases.
Tyrosine kinase inhibitors (Rigosertib)	High-risk MDS after HMA failure	It is a Ras pathway inhibitor; It was suggested that MDS cases with primary HMA failure and those with high-risk benefited most from the treatment with rigosertib.
*FLT3* inhibitors (Gilteritinib, quizartinib)	MDS cases with FLT3 mutation	In high-risk MDS
*IDH* inhibitors (Ivosidenib, Enasidenib)	MDS with *IDH1* mutation MDS with *IDH2* mutation	In MDS after HMA failure
APR-246 or Eprenetapopt (*TP53* modulator)	MDS with *TP53* mutation	A new molecule that induces apoptosis of the p53 cancer cells by the reactivation of the mutant p53 protein through restoring the normal conformation; MDS patients with *TP53* are resistant to conventional chemotherapy; APR-246 showed significant activity in MDS cases with *TP53* mutation.
H3B-8800 (Spliseosome modulator)	MDS with spliceosome mutations	H3B-8800 is an oral molecule splicing modulator that preferentially destroys the cells with *SF3B1* gene mutation; The combination with HMA or luspatercept is under study.
Magrolimab	High-risk MDS (including those with *TP53* mutation)	It is a CD47 monoclonal antibody that functions as a macrophage checkpoint inhibitor; In combination with azacitidine, Magrolimab showed promising results in AML and in MDS cases with the *TP53* mutation

allo-SCT—allogeneic stem cell transplantation; LEN—Lenalidomide; del—deletion, ESA—Erythropoiesis stimulating agents HMA—Hypomethylating Agents; TGFβ—transforming growth factor β; IST—Immunosuppressive Therapy; OS—overall survival.

## Data Availability

Not applicable.

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
