# Peer review of "The Genetic Landscape of Myelodysplastic Neoplasm Progression to Acute Myeloid Leukemia"

_ijms, 2023, doi:10.3390/ijms24065734_

Round 1

Reviewer 1 Report

This manuscript, written by Dr. Banescu, review type, with the title of "The genetic landscape of progression from Myelodysplastic Neoplasm to Acute Myeloid Leukemia" makes a summary of the cytogenetics and molecular signatures of MDS, with focus on the transformation of AML. The mansucript is well written, it is easy to read, and to understand.

The myelodysplastic syndromes (MDS) comprise a group of hematologic malignancies characterized by clonal hematopoiesis, one or more cytopenias (ie, anemia, neutropenia, and/or thrombocytopenia), and abnormal cellular maturation. MDS shares clinical and pathologic features with acute myeloid leukemia (AML), but MDS has a lower percentage of blasts in peripheral blood and bone marrow (by definition, <20 percent). Patients with MDS are at risk for symptomatic anemia, infection, bleeding, and transformation to AML, the incidence of which varies widely across MDS subtypes.

Comments:

1) The classification of myelodysplastic neoplasms is changing. Could you please add more information in the introduction?

Zhang, Y., Wu, J., Qin, T. et al. Comparison of the revised 4th (2016) and 5th (2022) editions of the World Health Organization classification of myelodysplastic neoplasms. Leukemia 36, 2875–2882 (2022). https://doi.org/10.1038/s41375-022-01718-7

Khoury, J.D., Solary, E., Abla, O. et al. The 5th edition of the World Health Organization Classification of Haematolymphoid Tumours: Myeloid and Histiocytic/Dendritic Neoplasms. Leukemia 36, 1703–1719 (2022). https://doi.org/10.1038/s41375-022-01613-1

Please also compare with the International Consensus Classification:

Arber DA, Orazi A, Hasserjian RP, et al. International Consensus Classification of Myeloid Neoplasms and Acute Leukemias: integrating morphologic, clinical, and genomic data. Blood. 2022 Sep 15;140(11):1200-1228. doi: 10.1182/blood.2022015850. PMID: 35767897; PMCID: PMC9479031.

2) Could you please also highlight/expand the overview of the treatment of myelodysplastic syndromes?

3) Myelodysplastic neoplasms/syndromes (MDS) refer to a diverse group of hematologic malignancies with variable clinical courses and outcomes.

Regarding the prognosis. Is it feasible to expand or show a table of the different prognostic models?

IPSS-M (https://mds-risk-model.com/)

IPSS-R

PPM-MDS

4) In table 4. Is it possible to add a column with the function of the gene (oncogene, suppressor, function?)

Author Response

Attached for your consideration is a revised version of the manuscript “The genetic landscape of Myelodysplastic Neoplasm progression to Acute Myeloid Leukemia”,  ID ijms-2257470 which was submitted to International Journal of Molecular Sciences, section: Molecular Oncology, issue: Advanced Research in Acute Myeloid Leukemia.

Following the reviewer’s comments, we made the required modifications (written in red color) to the initial version of our manuscript, which we described point-by-point, as follows:

Reviewer 1

This manuscript, written by Dr. Banescu, review type, with the title of "The genetic landscape of progression from Myelodysplastic Neoplasm to Acute Myeloid Leukemia" makes a summary of the cytogenetics and molecular signatures of MDS, with focus on the transformation of AML. The mansucript is well written, it is easy to read, and to understand. The myelodysplastic syndromes (MDS) comprise a group of hematologic malignancies characterized by clonal hematopoiesis, one or more cytopenias (ie, anemia, neutropenia, and/or thrombocytopenia), and abnormal cellular maturation. MDS shares clinical and pathologic features with acute myeloid leukemia (AML), but MDS has a lower percentage of blasts in peripheral blood and bone marrow (by definition, <20 percent). Patients with MDS are at risk for symptomatic anemia, infection, bleeding, and transformation to AML, the incidence of which varies widely across MDS subtypes.

Comment 1: The classification of myelodysplastic neoplasms is changing. Could you please add more information in the introduction? Please also compare with the International Consensus Classification

Response: Thank you for the suggestions that improved the quality of the manuscript, also for the prompt and concise evaluation. More information was added in the Introduction section, also the articles indicated were discussed. We discussed, the 4th, 5th WHO, and ICC classifications, and also compared these classifications. WHO classification was discussed in accordance with ICC as recommended.

Comment 2: Could you please also highlight/expand the overview of the treatment of myelodysplastic syndromes?

Response: We expanded the overview of the treatment of myelodysplastic syndromes as it was suggested.

Comment 3: Myelodysplastic neoplasms/syndromes (MDS) refer to a diverse group of hematologic malignancies with variable clinical courses and outcomes. Regarding the prognosis. Is it feasible to expand or show a table of the different prognostic models? IPSS-M (https://mds-risk-model.com/), IPSS-R, PPM-MDS.

Response: Table 1 was added. Table 1 presents the characteristics of IPSS, IPSS-R, IPSS-M and PPM-MDS prognostic scoring systems in MDS. Thank you for your suggestion

Comment 4: In table 4. Is it possible to add a column with the function of the gene (oncogene, suppressor function?)

Response: In the table entitled “Gene mutation in MDS and in progression to leukemia” we added a column that mentions the function of the gene.

Reviewer 2

The manuscript was prepared very well. However, there are some concerns, in part important, so the review articles need revision, see below.

Response: Thank you to the reviewer for the suggestion and for prompt evaluation.

General comments

Comments 1. Introduction. Why is this study considered relevant? Why is this study necessary?

Response: In the present work, we aimed to summarize the discoveries that help the understanding of the MDS, the state-of-the-art of diagnosis, risk stratification, prognostic scoring systems, and the risk-adapted treatment in order to improve the survival of affected patients and prevention of MDS transformation into AML in MDS cases. Considering that MDS is characterized by a high risk of transformation into leukemia it is essential to know the genetic anomalies that help clinicians for better clinical management. In this narrative review, we have discussed the main evidence taking into consideration the recent international recommendation, clinical trials, original research, reviews, our experience, and real-world evidence.

Comments 2. Materials and Methods It should include some methodology, what databases you have used, what MESH terms. even if it is a narrative review, it must be known where the information described in the results has been contained.

Response: We added the information requested. In the present work, we aimed to summarize the discoveries that help the understanding of the MDS, the state-of-the-art of diagnosis, risk stratification, prognostic scoring systems, and the risk-adapted treatment in order to improve the survival of affected patients and prevention of MDS transformation into AML in MDS cases. In this narrative review, we have discussed the main evidence taking into consideration the recent international recommendation, clinical trials, original research, reviews, our experience, and real-world evidence. The studies cited are indexed in PubMed. Examples of keywords used are myelodysplastic neoplasm, myelodysplastic syndrome, MDS progression, target treatment, genetic of MDS,  and leukemic transformation of MDS.

Comments 3. Results / Discussion This is the strong part of the study; I congratulate the authors. Include a section on limitations and strengths. What does this article contribute to, the authors should make their own assessment and include their own discussion of the results shown in the manuscript?

Response: Limitations and strengths were added as indicated. The manuscript summarizes the discoveries that help the understanding of the MDS, the state-of-the-art of diagnosis, risk stratification, prognostic scoring systems, and the risk-adapted treatment in order to improve the survival of affected patients and prevention of MDS transformation into AML in MDS cases.

The discussions were performed as indicated.

Comments 4. the figure should be improved, the table should contain a legend.

Response: The figure was changed, and the legend was added to each table.

Comments 5. Explain why you select the RNA Splicing Mutations you describe?

Response The presence of mutations that interest the spliceosome genes (eg, SF3B1, SRSF2, U2AF1 genes), suggests AML progression from MDS, even in patients with a negative history of MDS diagnosis. In MDS, spliceosome mutations occur commonly in SRSF2, SF3B1, ZRSR2, and U2AF1 genes, while SF3A1, SF1, and ZXRSR2 gene mutations are rare with a frequency of 1% for each of them.

Comments 6. In the Conclusion section, state the most important outcome of your work. Do not simply summarize the points already made in the body — instead, interpret your findings at a higher level of abstraction. Show whether, or to what extent, you have succeeded in addressing the need stated in the Introduction (or objectives).

Response: Thank you. We performed the changes in the manuscript.

In addition, the entire manuscript was carefully evaluated for the English language.

Thus, by this letter and by the attached revised version of our manuscript we hope to have fulfilled all the observations and recommendations made by the Reviewers.

Thank you for your time and consideration.

On behalf of all authors of this work,

Yours sincerely,

Claudia Banescu

Reviewer 2 Report

The manuscript was prepared very well. However, there are some concerns, in part important, so the review articles need revision, see below.

General comments

Introduction

-        Why is this study considered relevant?

-        why is this study necessary?

Materials and Methods

-        It should include some methodology, what databases you have used, what MESH terms. even if it is a narrative review, it must be known where the information described in the results has been contained.

Results / Discussion

·       This is the strong part of the study; I congratulate the authors.

·       Include a section on limitations and strengths.

·       What does this article contribute to, the authors should make their own assessment and include their own discussion of the results shown in the manuscript?

·       the figure should be improved, the table should contain a legend

·       Explain why you select the RNA Splicing Mutations you describe?

·       In the Conclusion section, state the most important outcome of your work. Do not simply summarize the points already made in the body — instead, interpret your findings at a higher level of abstraction. Show whether, or to what extent, you have succeeded in addressing the need stated in the Introduction (or objectives).

Author Response

(The authors gave the same response as above.)

Round 2

Reviewer 2 Report

The manuscript has gained in quality with the changes; however, a better organization of the different sections would be advisable.

Table 1 has too much text, could it be improved? there are some abbreviations not included in the table footer

include some more figures in the sections that do not have it

You must consist of what your manuscript contributes from the point of view of the authors

Author Response

Attached for your consideration is a re-revised version of the manuscript “The genetic landscape of Myelodysplastic Neoplasm progression to Acute Myeloid Leukemia”,  ID ijms-2257470 which was submitted to International Journal of Molecular Sciences, section: Molecular Oncology, issue: Advanced Research in Acute Myeloid Leukemia.

Following the reviewer’s comments, we made the required modifications (written in blue color) to the initial version of our manuscript, which we described point-by-point, as follows:

Comment: The manuscript has gained in quality with the changes; however, a better organization of the different sections would be advisable.

Response. Section 4 was organized into 2 subparts.

Table 1 has too much text, could it be improved? there are some abbreviations not included in the table footer:

Response The abbreviations were added to table 1, and also to the other tables, the text from Table 1 was reduced as was recommended.

Comment: include some more figures in the sections that do not have it

Response: two new figures were added

Comment: You must consist of what your manuscript contributes from the point of view of the authors

Response: We added the contribution of the manuscript.  

Thus, by this letter and by the attached re-revised version of our manuscript we hope to have fulfilled all the observations and recommendations made by the Reviewer.

Thank you for your time and consideration.

On behalf of all authors of this work,

Yours sincerely,

Claudia Banescu
